# Evaluation of Potential Anti-Hepatitis A Virus 3C Protease Inhibitors Using Molecular Docking

**DOI:** 10.3390/ijms23116044

**Published:** 2022-05-27

**Authors:** Reina Sasaki-Tanaka, Kalyan C. Nagulapalli Venkata, Hiroaki Okamoto, Mitsuhiko Moriyama, Tatsuo Kanda

**Affiliations:** 1Division of Gastroenterology and Hepatology, Department of Medicine, Nihon University School of Medicine, 30-1 Oyaguchi-kamicho, Itabashi-ku, Tokyo 173-8610, Japan; moriyama.mitsuhiko@nihon-u.ac.jp (M.M.); kanda.tatsuo@nihon-u.ac.jp (T.K.); 2Department of Pharmaceutical and Administrative Sciences, Saint Louis College of Pharmacy, University of Health Sciences and Pharmacy, St. Louis, MO 63010, USA; kalyan.venkata@uhsp.edu; 3Division of Virology, Department of Infection and Immunity, Jichi Medical University School of Medicine, 3311-1 Yakushiji, Shimotsuke-shi, Tochigi 329-0498, Japan; hokamoto@jichi.ac.jp

**Keywords:** 3C protease, HAV, in silico screening, molecular docking, protease inhibitors

## Abstract

Hepatitis A virus (HAV) infection is a major cause of acute hepatitis worldwide and occasionally causes acute liver failure and can lead to death in the absence of liver transplantation. Although HAV vaccination is available, the prevalence of HAV vaccination is not adequate in some countries. Additionally, the improvements in public health reduced our immunity to HAV infection. These situations motivated us to develop potentially new anti-HAV therapeutic options. We carried out the in silico screening of anti-HAV compounds targeting the 3C protease enzyme using the Schrodinger Modeling software from the antiviral library of 25,000 compounds to evaluate anti-HAV 3C protease inhibitors. Additionally, in vitro studies were introduced to examine the inhibitory effects of HAV subgenomic replicon replication and HAV HA11-1299 genotype IIIA replication in hepatoma cell lines using luciferase assays and real-time RT-PCR. In silico studies enabled us to identify five lead candidates with optimal binding interactions in the active site of the target HAV 3C protease using the Schrodinger Glide program. In vitro studies substantiated our hypothesis from in silico findings. One of our lead compounds, Z10325150, showed 47% inhibitory effects on HAV genotype IB subgenomic replicon replication and 36% inhibitory effects on HAV genotype IIIA HA11-1299 replication in human hepatoma cell lines, with no cytotoxic effects at concentrations of 100 μg/mL. The effects of the combination therapy of Z10325150 and RNA-dependent RNA polymerase inhibitor, favipiravir on HAV genotype IB HM175 subgenomic replicon replication and HAV genotype IIIA HA11-1299 replication showed 64% and 48% inhibitory effects of HAV subgenomic replicon and HAV replication, respectively. We identified the HAV 3C protease inhibitor Z10325150 through in silico screening and confirmed the HAV replication inhibitory activity in human hepatocytes. Z10325150 may offer the potential for a useful HAV inhibitor in severe hepatitis A.

## 1. Introduction

In general, hepatitis A, which is caused by hepatitis A virus (HAV) infection, is a self-limited disease, but it occasionally causes acute liver failure, which can be fatal for some patients or can require liver transplantation for survival. Although effective vaccines for HAV infection have already been developed, there are several medical problems. HAV vaccination rates are still low in some countries, due to their high costs, vaccine supply shortages, etc. [1].

It has been identified that several vaccinated patients in the demographic of men who have sex with men (MSM) are affected by hepatitis A outbreaks [2]. Next-generation sequencing techniques (NGSs) revealed the emergence of HAV antigenic variants in and around the main epitopes of the capsids, suggesting the positive selection of antigenic variants in some vaccinated patients [3]. Therefore, new, effective treatment should be developed for the treatment of HAV infection.

HAV has a positive-polarity, single-stranded ~7.6 kb genome [4]. The 5′ untranslated region (UTR) is 734–740 nucleotides in length and translation occurs in a cap-independent fashion under the control of an internal ribosomal entry site (IRES) thought to be mainly located in the 5′UTR [5,6]. The 3′UTR, which is 40–80 nucleotides in length, includes a translation terminator sequence with a poly(A) tail [4]. The remainder of the genome is composed of a single open reading frame with three distinct regions (P1, P2, and P3) and is translated as a single polyprotein with ~2225–2227 amino acids in length. HAV 3C protease cleaves this polyprotein into at least four capsid proteins (VP4, VP2, VP3, and VP1) and seven nonstructural proteins (2A, 2B, 2C, 3A, 3B, 3C, and RNA-dependent RNA polymerase 3D) [7]. HAV 3C also inhibits HAV IRES-dependent translation and the polypyrimidine tract-binding protein (PTB) to give way to subsequent HAV genome replication [8]. Thus, as HAV 3C protein plays an important role, it is one of the attractive targets of anti-HAV drugs [9,10,11,12].

The discovery of small molecules with biological activities is important to discover drug candidates as potential therapeutic agents and investigate biological mechanisms [13]. Since traditional high-throughput screening has high cost and time requirements, it is useful to use computational methods to select candidate ligands prior to experimental testing [14]. The HAV 3C structural information is determined via X-ray crystallography. The first step in this process is to identify molecules that bind to a HAV 3C protease using a molecular docking study by integrating a virtual screening and computational approach with the chemical synthesis of selected molecules. The molecular docking study positions the three-dimensional models of small molecules into a HAV 3C binding pocket, and associated scoring functions subsequently predict the binding affinities of these candidate molecules [15]. However, molecular docking studies have some inherent inaccuracies [16].

Thus, to confirm the anti-HAV ability of candidate molecules, we evaluated the cytotoxicity and anti-HAV activity in human hepatoma cell lines in the next step. In the present study, we examined the effects of five candidate molecules on HAV HM175 genotype IB subgenomic replicon replication and HAV HA11-1299 genotype IIIA replication in cell culture infection systems.

## 2. Results

### 2.1. Molecular Docking Study and Identification of Five Ligands for HAV 3C Protease

We ran the screening and docking of potential small molecule inhibitors in the crystal structure of HAV 3C protease (PDB ID 2CXV) utilizing the same binding pocket that was previously reported (Figure 1A–D) [12].

The Enamine (Kyiv, Ukraine) library was used for our docking analysis. This is not a repurposing study; the ligands used are novel molecules. The library used in this project is a specialized antiviral library consisting of 25,000 molecules. The library selection was based on its preference for viral targets. These compounds from the antiviral library were screened against the binding pocket of HAV 3C protease using the Schrodinger Glide program.

Finally, five chemical compounds, Z2351109846, Z10325150, Z1452073950, Z287374370, and Z1208291016, which exhibited potential interactions with the key amino acids in the binding site of HAV 3C protease enzymes, were identified and selected for the present study (Table 1). Another key reason for their selection was based on their chemical diversity and ease of synthesis. The structures of these compounds are shown in Figure 1E.

### 2.2. Effects of Five Molecules on Viability of HuhT7 Cells

We next examined the cytotoxic effects of five molecules on HuhT7 cells using dimethylthiazol carboxymethoxyphenyl sulfophenyl tetrazolium (MTS) assays [17]. Cell viability was compared to dimethyl sulfoxide (DMSO)-treated controls. As depicted in Figure 2A–E, five molecules did not impact the cell viability of HuhT7 cells at concentrations of 100 μg/mL in HuhT7 cells for 48 h.

### 2.3. Z2351109846, Z10325150, Z1452073950, and Z287374370 Significantly Inhibited HAV Subgenomic Replicon Replication in HuhT7 Cells

We evaluated the effects of Z2351109846, Z10325150, Z1452073950, Z287374370, and Z1208291016 on HAV subgenomic replicon replication in HuhT7 cells. Cells were transiently transfected with 0.2 μg of pT7-18f-LUC (HAV strain HM175 18f) using Effectene Transfection Reagent (Qiagen, Hilden, Germany) according to the manufacturer’s instructions [17,18], and after 24 h of transfection, cells were treated with 0, 1, 10, and 100 μg/mL of Z2351109846, Z10325150, Z1452073950, Z287374370, and Z1208291016, respectively. After 72 h of transfection, luciferase activity was measured using a luminometer. Z2351109846, Z10325150, Z1452073950, and Z287374370 treatment significantly inhibited HAV subgenomic replicon replication at a concentration of 100 μg/mL (Figure 3A–D). Thus, 100 μg/mL Z2351109846, Z10325150, Z1452073950, and Z287374370 effectively inhibited HAV subgenomic replicon replication.

### 2.4. Effects of Five Molecules on Viability of Huh7 Cells

Cytotoxic effects of five molecules for Huh7 cells were evaluated using MTS assays. Cell viability was compared to DMSO-treated controls. As depicted in Figure 4A–E, Z10325150, Z1452073950, Z287374370, and Z1208291016 did not impact cell viability at concentrations of 100 μg/mL in Huh7 cells for 72 h. Compared to DMSO-treated control cells, 50 μg/mL or higher of Z2351109846 significantly reduced cell viability (*p* < 0.01).

### 2.5. Z10325150 Significantly Downregulated HAV Replication in HAV-Infected Huh7 Cells

Next, we evaluated the effects of five molecules on HAV replication in Huh7 cells. We measured HAV RNA levels using real-time reverse transcription (RT)-polymerase chain reaction (PCR). Huh7 cells were infected with HAV HA11-1299 genotype IIIA and treated with Z2351109846 at 0 and 10 μg/mL; Z10325150, Z1452073950, and Z287374370 at 0, 50, and 100 μg/mL; and Z1208291016 at 0 and 100 μg/mL, respectively (Figure 5A–E). Ninety-six hours after infection, cellular RNA was extracted. As shown in Figure 4B, HAV RNA levels were significantly inhibited by the treatment of Z10325150 in HAV-infected Huh7 cells. These results show that 100 μg/mL Z10325150 results in a significant reduction in HAV RNA levels.

## 3. Discussion

In the present study, to identify potential drugs against HAV, we performed the in silico screening of anti-HAV drugs targeting 3C protease enzymes using molecular docking. Previous crystallographic studies of the HAV 3C protease active site have shown that a catalytic site formed by triad His44, Asp84, and Cys172, alongside oxyanion hole and neighboring amino acid residues, might be a useful target for protease-inhibitor interactions. We selected three amino acids (catalytic triad) to generate a grid in Schrodinger molecular modeling software for docking. The grid site generated was an approximate site for binding interactions with the ligands. We, therefore, postulated that a small molecule binding to the active site could potentially inhibit the protease activity. We employed the Schrodinger Glide molecular docking program to study the interactions between the ligand and HAV 3C protease enzyme at an atomic level [19].

We selected five compounds with the best fits in the active site (Figure 1 and Table 1). All five compounds selected have potentially strong binding interactions with residues in the pocket, and interestingly, the stereochemistry of the compounds played a significant role in binding. Our selected pharmacophores have great chemical diversity with varied functional groups and heterocyclic structures. Due to the ease of synthesis, we could commercially obtain them from Enamine chemicals.

Finally, we further validated the in silico data from our docking models with the in vitro anti-HAV efficacy of these molecules against the HAV 3C protease enzyme. The small molecule Z10325150 could inhibit HAV genotype IB subgenomic replicon replication and HAV genotype IIIA HA11-1299 replication in human hepatoma cell lines. Our in silico docking studies predict that the small molecule Z10325150 performs key binding interactions with HAV 3C protease. Most notably, the –OH group present in Z10325150 interacts with His145 in the active site (Figure 1D). We are delighted to see that the role hydroxyl group could potentially play in inhibiting HAV 3C protease resembles that of commercially available human immunodeficiency virus (HIV) protease inhibitors [20].

HAV infection is a major cause of acute hepatitis worldwide [21] and occasionally causes acute liver failure, leading to death without liver transplantation [22]. Although HAV vaccination is available, the prevalence of HAV vaccination is not sufficient in some countries [1]. The improvement in the public health environment also reduces the population with anti-HAV immunity [1,22,23]. As these situations could result in hepatitis A outbreaks, it is important to develop more specific anti-HAV drugs.

HAV 3C protease enzymes play a role in the processing of the HAV polyprotein and in the promotion of HAV replication and infection [24]. Protease inhibitors play an important role in the treatment of HIV type 1 (HIV-1) and hepatitis C virus (HCV) infection [25,26]. In the present study, compared with the untreated control (100%), Z10325150 monotherapy reduced 47% and 36% of HAV genotype IB HM175 subgenomic replicon replication in HuhT7 cells and HAV genotype IIIA HA11-1299 replication in Huh7 cells, respectively (Figure 3B and Figure 5B). Our results support the conception that HAV 3C protease is one of the attractive targets of direct-acting antivirals (DAAs) against HAV infection.

Notably, 4 of the 5 compounds suppressed HAV genotype IB HM175 subgenomic replicon replication in HuhT7 cells, although only Z10325150 suppressed HAV genotype IIIA HA11-1299 replication in Huh7 cells (Figure 3 and Figure 5). We used the X-ray crystal structure of HAV 3C protease, which was derived from HAV genotype IB HM175 [12]. Z10325150 could inhibit HAV replication and be effective for HAV genotypes IB and IIIA infection.

The docking score of Z10325150 was −7.488 kcal/mol (Table 1). In our previous work, we observed that the compounds with very high docking scores did not have any biological activity. Our selection of compounds was not merely based on docking scores, but we relied on the functional group interactions in the active site (Figure 1).

Computational studies of 3C proteases and other *Picornaviridae* family members may help us identify compounds that could potentially be broad-spectrum inhibitors for 3C proteases of other members of the *Picornaviridae* family [12]. Molecular docking techniques have been employed to study the inhibitory activity of other drugs in interaction with the HAV 3C protease [27,28,29]. In the cytopathic effect inhibition assays, amantadine and 3-benzyl(phenethyl)benzo[g]quinazolines derivatives were tested against the cytopathogenic HAV HM175 strain; the quinazoline derivatives were more potent against HAV than amantadine [29]. Interestingly, molecular docking studies show that quinazoline derivatives and amantadine have negative binding energies, inferring that the activity might be due to their HAV 3C proteinase inhibition process. However, we do not fully understand the mechanism of action of amantadine against HAV 3C [8]. 

Recently, we reported on the inhibitory effects of favipiravir on HAV replication [17]. We also examined the effects of the combination therapy of Z10325150 and favipiravir on HAV genotype IB HM175 subgenomic replicon replication and HAV genotype IIIA HA11-1299 replication. Compared with Z10325150 monotherapy (100%), this combination therapy reduced 80% and 67% of HAV genotype IB HM175 subgenomic replicon replication in HuhT7 cells and HAV genotype IIIA HA11-1299 replication in Huh7 cells, respectively.

The few limitations of our study are as follows: (1) Z10325150 might show relatively weaker inhibition of HAV replication as a DAA. Our goal is to develop inhibitors with stronger inhibition of HAV replication. (2) We do not have in vivo data and are currently in the process of performing in vivo studies to determine the safety and efficacy of these drugs in animal models. (3) In infections with other viruses, such as HIV and HCV, a combination therapy of protease inhibitors and other classes of antiviral drugs is generally recommended, and the combination of Z10325150 and several inhibitors other than Favipiravir are currently under examination. Our future goal is also to evaluate the efficacy of this candidate molecule, Z10325150, in patients coinfected with HAV and other viruses.

## 4. Materials and Methods

### 4.1. Software, Database, and Molecular Docking

In order to gain an insight into the binding interaction of the investigated compounds with HAV 3C protease enzymes, we performed molecular docking studies based on the crystal structures of HAV 3C protease by utilizing the X-ray crystal structure of HAV 3C protease (PDB ID: 2CXV) taken from the RCSB databank [12]. Probable binding sites were predicted near the catalytic triad His44, Asp84, and Cys172, and a grid was generated. The initial structure preparation of proteins and ligands required for docking and visual analyses at different stages were performed using Schrodinger Maestro (New York, NY, USA). The molecular docking simulations of ligands into the binding sites of HAV 3C protease were carried out using the Schrodinger Glide program. The strategy for identifying the best docked pose involved a Random Conformation Search, which utilizes the grid-based scoring functions of Coulombic and Lennard–Jones forces. Based on the Glide scores and binding interactions in the active site, the five best hits were selected for in vitro evaluation.

### 4.2. Cell Lines and Reagents

The human hepatoma cell lines Huh7 and HuhT7, the latter being a stably transformed derivative of Huh7 expressing T7 RNA polymerase, were used [18]. The Huh7 and HuhT7 cells were kindly provided by Prof. Bartenschlager and Prof. Gauss-Müller, respectively [18,30]. Cells were maintained in Roswell Park Memorial Institute medium (RPMI; Sigma-Aldrich, St. Louis, MO, USA) containing 10% heat-inactivated fetal bovine serum (FBS; Sigma-Aldrich), 100 units/mL penicillin, and 100 μg/mL streptomycin (Sigma-Aldrich) under a 5% CO2 atmosphere at 37 °C. HAV genotype IIIA HA11-1299 was used for HAV infection in the present study. Replication-competent HAV subgenomic replicon (HAV subgenomic replicon) pT7-18f-LUC contains an open-reading frame of firefly luciferase flanked by the first four amino acids of HAV polyprotein and by 12 C-terminal amino acids of VP1. These segments are followed by P2 and P3 domains of HAV polyprotein (HAV genotype IB HM175 18f) [18]. The five chemical compounds, Z2351109846, Z10325150, Z1452073950, Z287374370, and Z1208291016, which are ligands for HAV 3C protease enzymes, were selected and purchased from Enamine.

### 4.3. Cell Viability Assays

For the evaluation of cell growth and cell viability, MTS assays were performed using the CellTiter 96 Aqueous One-Solution cell proliferation assay (Promega, Madison, WI, USA). Enzyme activity was measured using a Bio-Rad iMark microplate reader (Bio-Rad, Hercules, CA, USA) at a 490 nm wavelength.

### 4.4. Transfection of HAV Subgenomic Replicon into HuhT7 Cells and Luciferase Assays

Twenty-four hours prior to transfection, approximately 1 × 10^5^ cells/well were seeded on a 24-well plate (AGC TECHNO GLASS, Tokyo, Japan). Cells were transfected with 0.2 μg pT7-18f-LUC (HAV strain HM175 18f) using Effectene Transfection Reagent (Qiagen) following the manufacturer’s protocol. After 24 h of transfection, the cells were treated with 0, 10, and 100 μg/mL of the five compounds. After 72 h of transfection, cells were harvested using reporter lysis buffer (Toyo Ink, Tokyo, Japan), and luciferase activity was determined using Luminescencer JNR II AB-2300 (ATTO, Tokyo, Japan).

### 4.5. Infection of HAV Genotype IIIA HA11-1299 into Huh7 Cells

HAV genotype IIIA HA11-1299 was inoculated for HAV infection in the present study [31]. Before 24 h of infection, Huh7 cells were seeded at a density of 3 × 105 cells/well on 6-well plates (AGC TECHNO GLASS). Cells were washed twice with phosphate-buffered saline (PBS) and infected with the HAV genotype IIIA HA11-1299 at a multiplicity of infection (MOI) of 0.1 in serum-free medium. HAV inoculum was incubated with hepatocytes for 6 h, and we added 1 mL of medium containing 2% FBS. After 24 h of incubation, cells were washed once with PBS, followed by the addition of 1 mL of RPMI containing 5% FBS. Then, several concentrations of drugs were added to HAV-infected Huh7 cells according to the results of cell viability assays. After 96 h of infection, cellular RNA was extracted using the RNeasy Mini Kit (Qiagen), and HAV RNA levels were determined using real-time RT-PCR.

### 4.6. RNA Extraction and Quantification of HAV RNA

Total cellular RNA was extracted from harvested cells using the RNeasy Mini Kit (Qiagen) according to the manufacturer’s instructions. cDNA was synthesized with oligo dT primers and random hexamers using the PrimeScript RT reagent kit (Perfect Real Time; Takara, Otsu, Japan). Reverse transcription was performed at 37 °C for 15 min, followed by 95 °C for 5 s. For HAV RNA quantification, the following primer set was used: sense primer 5′-AGGCTACGGGTGAAACCTCTTAG-3′, antisense primer 5′-GCCGCTGTTACCCTATCCAA-3′ [32] and actin quantified with sense primer 5′-CAGCCATGTACGTTGCTATCCAGG-3′ and antisense primer 5′-AGGTCCAGACGCAGGATGGCATG-3′ [17]. Real-time PCR was performed using the Power SYBR Green Master Mix (Thermo Fisher Scientific, Tokyo, Japan) with a 7500 Fast real-time PCR system (Applied Biosystems). PCR reaction was performed as follows: 95 °C for 10 min, followed by 40 cycles of 95 °C for 15 s and 60 °C for 1 min. The housekeeping gene actin was used for normalization, and data were analyzed using the comparative threshold cycle method. The relative quantification of gene expression using the ddCt method correlated with absolute gene quantification obtained by the standard curve.

### 4.7. Statistical Analysis

Data are expressed as means ± standard deviations (SD). Statistical analysis was performed using Student’s *t*-test. *p* < 0.05 was considered significant. All experiments were performed in triplicate.

## 5. Conclusions

We evaluated a potent HAV 3C protease inhibitor with activity against HAV genotype IB subgenomic replicon-transfected and HAV genotype IIIA HA11-1299-infected human hepatoma cells [27,28,29]. In particular, we showed that the presence of the hydroxyl group is important to inhibit the protease activity. Overall, this study suggests that small molecule inhibitors binding to the active site might be a promising approach for the development and optimization of new anti-HAV 3C protease inhibitors.

## Figures and Tables

**Figure 1 ijms-23-06044-f001:**
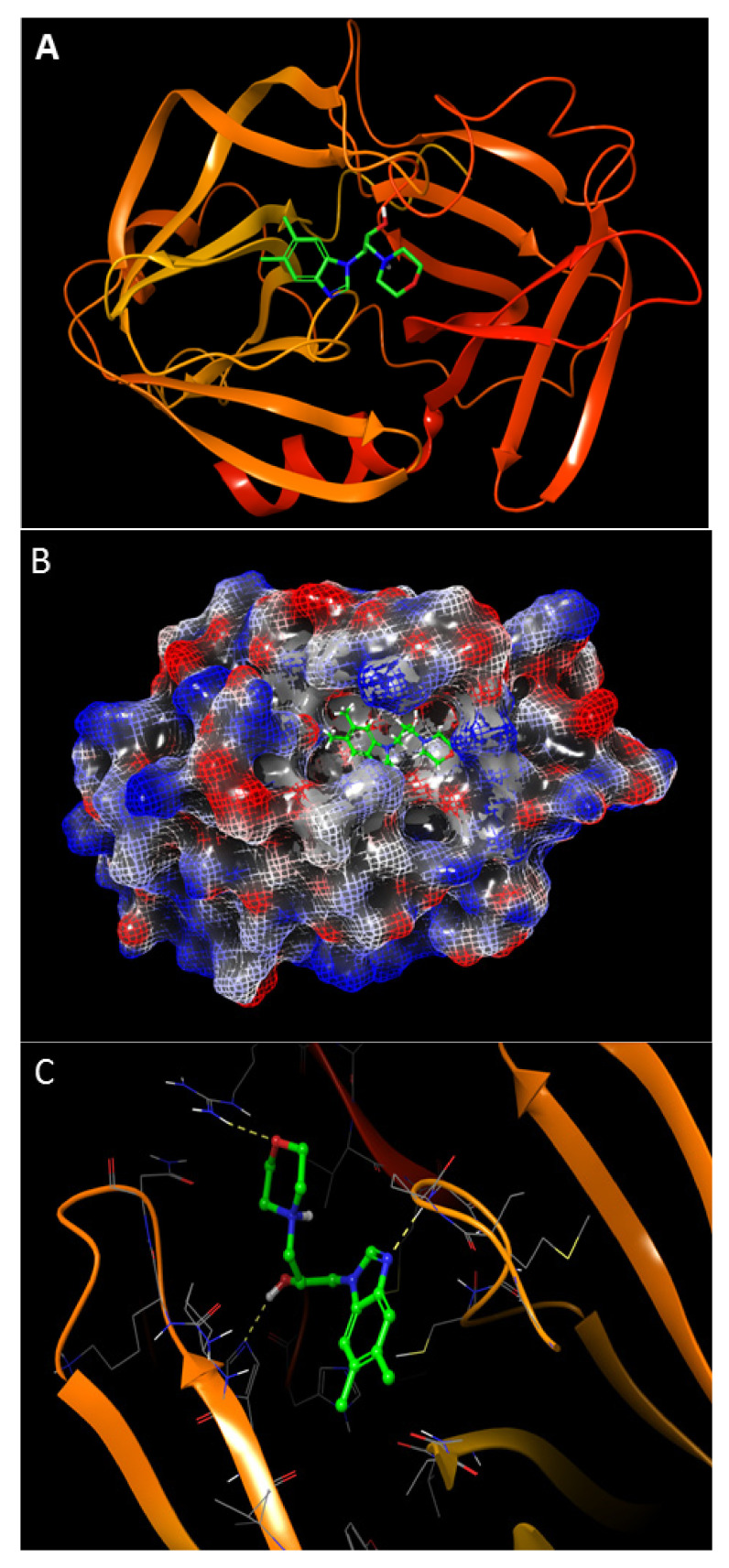
Molecular docking study and the structure of ligands for hepatitis A virus (HAV) 3C protease in the present study: (**A**) docking of Z10325150 in the active site of HAV 3C protease; (**B**) electrostatic potential of HAV 3C protease in complex with Z10325150; (**C**,**D**) 3D and 2D interactions between Z10325150 and HAV 3C protease; (**E**) structure of ligands for HAV 3C protein.

**Figure 2 ijms-23-06044-f002:**
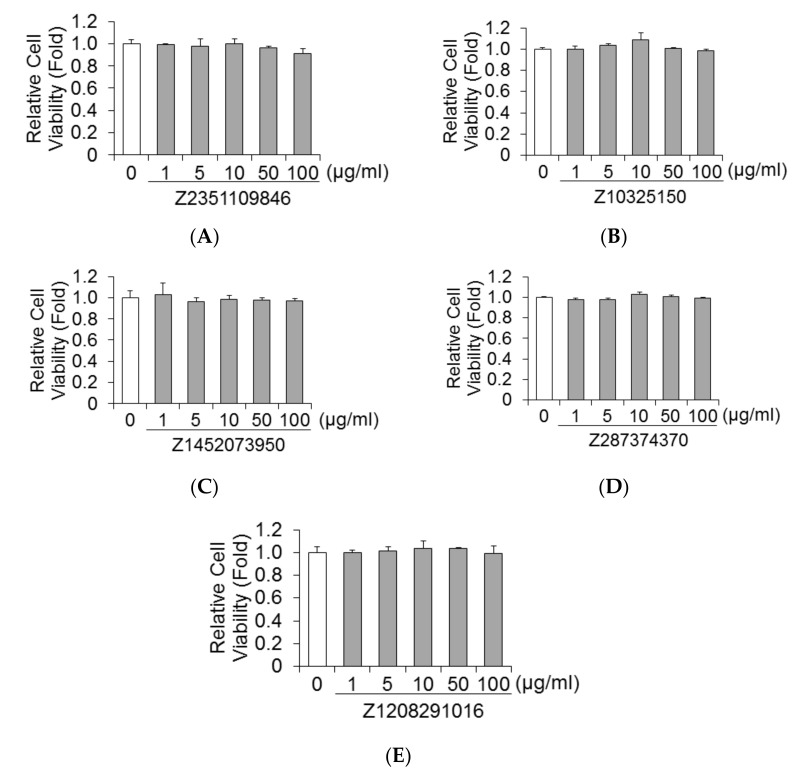
Effects of Z2351109846, Z10325150, Z1452073950, Z287374370, and Z1208291016 on cell viabilities of HuhT7 cells. Cell viabilities of HuhT7 cells treated with Z2351109846 (**A**), Z10325150 (**B**), Z1452073950 (**C**), Z287374370 (**D**), and Z1208291016 (**E**). HuhT7 cells were treated with each drug for 48 h. Cell viabilities were determined via dimethylthiazol carboxymethoxyphenyl sulfophenyl tetrazolium (MTS) assay. Data are expressed as means and standard deviations of triplicate determinations from three independent experiments. Statistical significance was analyzed using the two-tailed Student’s *t*-test.

**Figure 3 ijms-23-06044-f003:**
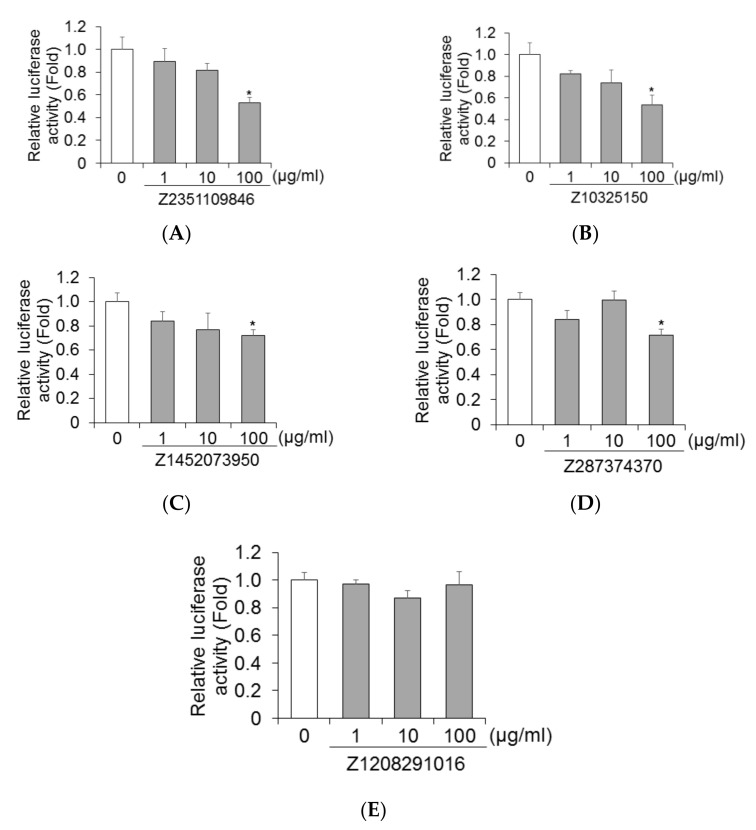
Z2351109846, Z10325150, Z1452073950, and Z287374370 inhibited HAV genotype IB subgenomic replicon replication. HuhT7 cells transfected with the HAV genotype IB subgenomic replicon. After 24 h of transfection, cells were treated with Z2351109846 (**A**), Z10325150 (**B**), Z1452073950 (**C**), Z287374370 (**D**), and Z1208291016 (**E**) for 48 h, and luciferase activity was determined after 72 h of transfection. Data are presented as the means and standard deviations of triplicate determinations from at least three independent experiments. Statistical significance was analyzed using the two-tailed Student’s *t*-test: * *p* < 0.05.

**Figure 4 ijms-23-06044-f004:**
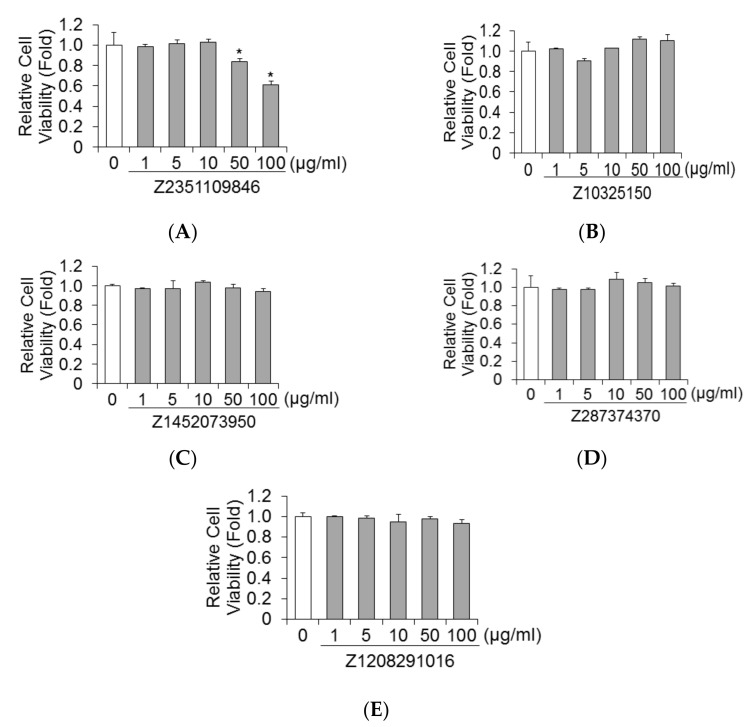
Effects of Z2351109846, Z10325150, Z1452073950, Z287374370, and Z1208291016 on cell viabilities of Huh7 cells. Cell viabilities of Huh7 cells treated with Z2351109846 (**A**), Z10325150 (**B**), Z1452073950 (**C**), Z287374370 (**D**), and Z1208291016 (**E**). Huh7 cells were treated with each drug for 72h. Cell viabilities were determined by dimethylthiazol carboxymethoxyphenyl sulfophenyl tetrazolium (MTS) assay. Data are expressed as means and standard deviations of triplicate determinations from three independent experiments. Statistical significance was analyzed using the two-tailed Student’s *t*-test: * *p* < 0.05.

**Figure 5 ijms-23-06044-f005:**
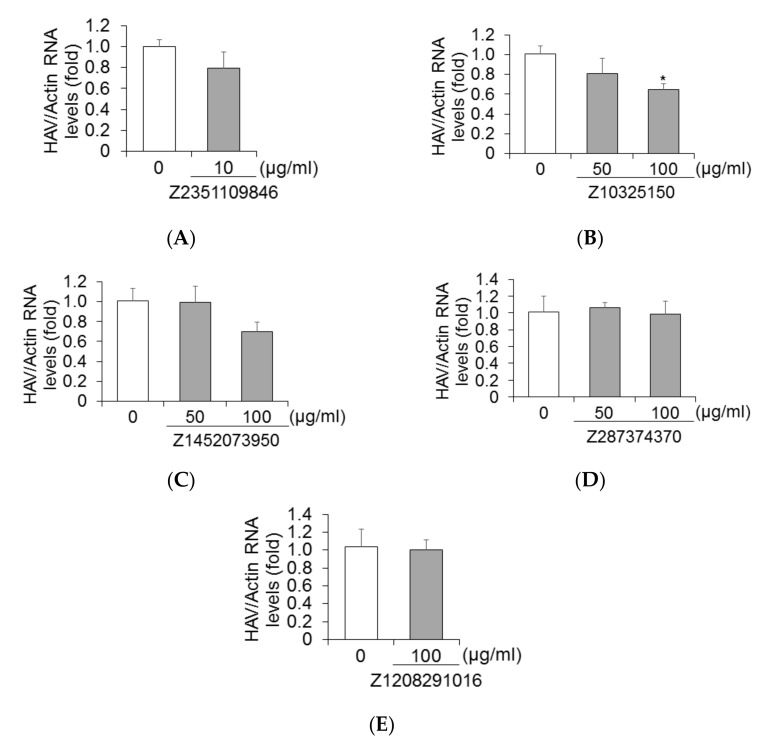
Z10325150 could inhibit HAV genotype IIIA HA11-1299 replication. Huh7 cells infected with HAV genotype IIIA HA11-1299 were treated with Z2351109846 (**A**), Z10325150 (**B**), Z1452073950 (**C**), Z287374370 (**D**), and Z1208291016 (**E**) for 72 h. HAV RNA levels were measured using real-time RT-PCR. Actin mRNA was used as an internal control. Data are presented as the means and standard deviations of triplicate determinations from at least three independent experiments. Statistical significance was analyzed using the two-tailed Student’s *t*-test: * *p* < 0.05.

**Table 1 ijms-23-06044-t001:** Ligands for hepatitis A virus 3C protease used in the present study.

Compound Name	Glide Scores (kcal/mol)	Molecular Weight	Formula
Z2351109846	−7.407	300.31	C_15_H_16_N_4_O_3_
Z10325150	−7.488	289.37	C_16_H_23_N_3_O_2_
Z1452073950	−7.364	317.43	C_18_H_27_N_3_O_2_
Z287374370	−7.362	313.33	C_17_H_16_FN_3_O_2_
Z1208291016	−7.232	279.31	C_14_H_18_FN_3_O_2_

## Data Availability

The data underlying this article are available in this article.

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
