# Peer review of "Evaluation of Potential Anti-Hepatitis A Virus 3C Protease Inhibitors Using Molecular Docking"

_ijms, 2022, doi:10.3390/ijms23116044_

Round 1
Reviewer 1 Report
A computational approach is undertaken to screen a publicly available data bank for ligands interacting with 3C protease enzyme of hepatitis A virus. The modeling software was used for the screening and identification of five potential ligands. The five ligands were further screened using cell based in vitro essay singling one ligand as a potential antiviral.
Some points need to be addressed:
The manuscript would benefit from some additional information being provided:
What kind of library is used? Is this a repurposing approach? Why is the number of ligands limited to about 25.000? Have they been chosen by some chemical/pharmaceutical criteria?
So far, the ‘design’ aspect as proposed in the title is not clearly elaborated. Have the molecules be chemically modified to improve the data?
Figure 1D: the resolution of the figure is not sufficient.
Minor points
Page 7, line 181: Picornaviridae
Author Response
Response to Reviewer 1 Comments
Thank you very much for your encouraging comments.
Response to your comments: “What kind of library is used? Is this a repurposing approach? Why is the number of ligands limited to about 25.000? Have they been chosen by some chemical/pharmaceutical criteria?”
Thank you for your valuable comments. We greatly appreciate the reviewer’s critical analysis and insight into the study. We utilized Enamine library for our docking analysis. It is not a repurposing study; the ligands used are novel molecules. The library used in this project is a specialized anti-viral library consisting of 25,000 molecules. Yes, the library selection was based on its preference for viral targets.
According to your suggestions, we also revised our manuscript as follows.
In Results section,
2.1. Molecular Docking Study and Identification of Five Ligands for HAV 3C Protease
We ran the screening and docking of potential small molecule inhibitors in the crystal structure of HAV 3C protease (PDB ID 2CXV) utilizing the same binding pocket that was previously reported (Figure 1A–D) [12].
Enamine (Kyiv, Ukraine) library were used for our docking analysis. It is not a repurposing study; the ligands used are novel molecules. The library used in this project is a specialized antiviral library consisting of 25,000 molecules. The library selection was based on its preference for viral targets. These compounds from antiviral library were screened against the binding pocket of HAV 3C protease using Schrodinger Glide program.…
Response to your comments: “So far, the ‘design’ aspect as proposed in the title is not clearly elaborated. Have the molecules be chemically modified to improve the data?”
Thank you for your valuable comments. The reviewer poses an excellent and thought-provoking question. We agree with the reviewer, we have not yet carried out any specific design changes to the molecules in this study. Our future goal is to make changes to the molecule to improve its potency. We dropped the ‘design’ component from the title.
Response to your comments: “Figure 1D: the resolution of the figure is not sufficient.”
Thank you for your valuable comments. We are grateful to the reviewer for this excellent suggestion. Accordingly, we revised the Figure 1D to make it look better.
Response to your comments: “Page 7, line 181: Picornaviridae”
Thank you for your valuable comments. We revised our manuscript accordingly.

Reviewer 2 Report
The authors performed glide docking to screening antiviral compounds target HAV 3C protease. IN vitro assays were then performed to evaluate those compounds’ antiviral activities in human hepatocytes. The best compound, Z10325150, can inhibit viral replication in two Human hepatoma cell lines, Huh7 and HuhT7, with weak activities (3-4 micromole). The paper may not be publishable without adequately addressing the following concerns.
- The docking scores around -7.3 kcal/mol are pretty low, and it is hard to rank them as those docking scores are very close.
- Covalent-binding with C172 was involved in the molecular mechanism of inhibition as reported by Jin et al (J. Mol Biol, 354, 854-871), is there any covalent bond formed between Z10325150 and C172?
- The authors did not perform binding assays or resolved experimental structures to confirm the compound truly interacts with the 3C protease. Thus, the sentence “Z10325150 makes key binding interactions in the HAV 3C protease active site.” in page 7 is not adequate.
Other minor issues:
- The quality of Figure 1 (especially 1D) is very low. The fonts are too small and the interacting residues were not labeled.
- Figure 5B is smaller than pictures in other panels
Author Response
Response to Reviewer 2 Comments
Thank you very much for your valuable comments.
Response to your comments: “The docking scores around -7.3 kcal/mol are pretty low, and it is hard to rank them as those docking scores are very close.”
Thank you for your valuable comments. The reviewer poses an excellent and thought-provoking question regarding docking score. From our earlier experience, we have seen that the compounds with very high docking scores did not have any biological activity. Our selection of compounds was not merely based on docking scores but we relied on the functional group interactions in the active site. In addition, we did not rank these compounds by docking scores as shown in Table 1.
According to your suggestions, we also revised our manuscript as follows.
In Discussion section,
…effective for HAV genotypes IB and IIIA infection.
The docking score of Z10325150 is -7.488 kcal/mol (Table 1). From our earlier experience, we have seen that the compounds with very high docking scores did not have any biological activity. Our selection of compounds was not merely based on docking scores but we relied on the functional group interactions in the active site (Figure 1). Computational studies of 3C…
Response to your comments: “Covalent-binding with C172 was involved in the molecular mechanism of inhibition as reported by Jin et al (J. Mol Biol, 354, 854-871), is there any covalent bond formed between Z10325150 and C172?”
Thank you for your valuable comments. We greatly appreciate the reviewer’s critical analysis and insight into the mechanism. No, our compound does not make covalent bond interaction with Cysteine 172 as suggested in the literature. As you can see in the revised figure, 1D that compound does not interact with cys172. However, our future goal is tethering a functional group on to our active molecule, Z10325150 that can covalently interact with cys172. We really appreciate your suggestion.
Response to your comments: ”The authors did not perform binding assays or resolved experimental structures to confirm the compound truly interacts with the 3C protease. Thus, the sentence “Z10325150 makes key binding interactions in the HAV 3C protease active site.” in page 7 is not adequate.”
Thank you for your valuable comments. We greatly appreciate reviewer’s constructive feedback. We changed the sentence to “Our in silico docking studies predict that the small molecule Z10325150 makes key binding interactions with the HAV 3C protease.” in “Discussion section”.
Response to your minor comments: ”The quality of Figure 1 (especially 1D) is very low. The fonts are too small and the interacting residues were not labeled.
Figure 5B is smaller than pictures in other panels.”
Thank you for your valuable comments.
We are grateful to the reviewer for this excellent suggestion. Accordingly, we revised the Figure 1D to make it look better. We also revised Figure 5B.

Reviewer 3 Report
The author performed the in silico screening of anti-HAV compounds targeting 3C protease enzyme using Schrodinger Modeling software from the antiviral library of compounds to design and evaluate anti-HAV 3C protease inhibitors. One of our lead compounds, Z10325150, could inhibit HAV genotype IB subgenomic replicon replication and HAV genotype IIIA HA11-1299 replication in human hepatoma cell lines with no toxic effects. In conclusion, we identified HAV 3C protease inhibitor Z10325150 through in silico screening and confirmed the HAV replication inhibitory activity in human hepatocytes. However, your article is inadequately presented. Furthermore, there are many grammatical mistakes and spelling mistakes as well. Although the article has scientific rigor, several major flows need to be corrected before publication.
- Abstract and Introduction are quite premature and shallow. Ideally, readers expect to have a very brief account of the aims, methods, key findings, and conclusions of a study from an abstract with a couple of sentences from each part.
- The flow of the introduction is not complete and unspecific. My recommendation is to construct the sentences more lucid and legible for more productive comprehension.
- Author explores the binding affinity of HAV 3C protease using the docking method but they have reported the gatekeeper residue for this study.
- Relocking is not reflected in the manuscript so, the author should validate our docking model through redock approaches.
- Author should perform molecular dynamics for better stability and explore in detail the impact of fluctuation of RMSF during MD.
- Figure quality is not good. The author should change all figures in high resolution.
- Additionally, I am confused about the discussion. It does not seem to really discuss the data that was described in the manuscript. I would suggest that the authors refocus their discussion to clarify how the results of their work fit into the larger picture of what is current today instead of describing more of a literature background
Author Response
Response to Reviewer 3 Comments
Thank you very much for your valuable comments.
Response to your comments 1: “Abstract and Introduction are quite premature and shallow. Ideally, readers expect to have a very brief account of the aims, methods, key findings, and conclusions of a study from an abstract with a couple of sentences from each part.”
Thank you for your valuable comments. According to your suggestions, we extensively revised the Abstract section of our manuscript.
Response to your comments 2: “The flow of the introduction is not complete and unspecific. My recommendation is to construct the sentences more lucid and legible for more productive comprehension.”
Thank you for your valuable comments. According to your suggestions, we extensively revised the Introduction section of our manuscript.
Response to your comments 3: “Author explores the binding affinity of HAV 3C protease using the docking method but they have reported the gatekeeper residue for this study.
Thank you for your valuable comments. We greatly appreciate the reviewer’s critical analysis. In this study, we performed in silico screening of anti-HAV drugs targeting 3C protease. We selected three amino acids (catalytic triad) for generating a grid in Schrodinger molecular modeling software for docking. The grid site generated was an approximate site for binding interactions with the ligands.
Response to your comments 4 & 5: “Relocking is not reflected in the manuscript so, the author should validate our docking model through redock approaches. Author should perform molecular dynamics for better stability and explore in detail the impact of fluctuation of RMSF during MD.”
Thank you for your valuable comments. We thank the reviewer for the excellent suggestion. We would like to bring to reviewer’s notice that this study is at its initial stages, in our future research, we plan to modify the structure further and, in those studies, we plan to conduct MD studies to further understand the stability. In addition, due to lack of Linux facility at the current time we are unable to perform the MD calculations.
Response to your comments 6: “Figure quality is not good. The author should change all figures in high resolution.”
Thank you for your valuable comments. We are grateful to the reviewer for this excellent suggestion. Accordingly, we revised all the figures in the manuscript.
Response to your comments 7: “Additionally, I am confused about the discussion. It does not seem to really discuss the data that was described in the manuscript. I would suggest that the authors refocus their discussion to clarify how the results of their work fit into the larger picture of what is current today instead of describing more of a literature background.
Thank you for your valuable comments. Accordingly, we extensively revised the discussion section of our revised manuscript.

Round 2
Reviewer 2 Report
The authors have adequately addressed my concerns in the revised version. Thus, I am pleased to support the publication of this paper after necessary grammar check and revision to improve its readability.
Author Response
Response to Reviewer 2 Comments
Thank you very much for your encouraging comments.
Response to your comments: “The authors have adequately addressed my concerns in the revised version. Thus, I am pleased to support the publication of this paper after necessary grammar check and revision to improve its readability.”
Thank you for your valuable comments. We asked English editor of MDPI to edit our revised manuscript again and to improve our English. Please see “English-Editing-Certificate-44916”.

Reviewer 3 Report
The revised version of the manuscript includes all remarks and modifications indicated. The main concerns of the manuscript have been solved. In my opinion, the provided version is now suitable for publication
Author Response
Response to Reviewer 3 Comments
Thank you very much for your encouraging comments.
